# Doctors’ Professional and Personal Reflections: A Qualitative Exploration of Physicians’ Views and Coping during the COVID-19 Pandemic

**DOI:** 10.3390/ijerph20075259

**Published:** 2023-03-24

**Authors:** G. Camelia Adams, Monique Reboe-Benjamin, Mariam Alaverdashvili, Thuy Le, Stephen Adams

**Affiliations:** Department of Psychiatry, College of Medicine, Royal University Hospital, University of Saskatchewan, Ellis Hall, Saskatoon, SK S7N 0W8, Canada

**Keywords:** pandemic, COVID-19, physicians, mental health, coping, emotions

## Abstract

Numerous studies have examined the risks for anxiety and depression experienced by physicians during the COVID-19 pandemic. Still, qualitative studies investigating physicians’ views, and their discovered strengths, are lacking. Our research fills this gap by exploring professional and personal reflections developed by physicians from various specialties during the pandemic. Semi-structured interviews were conducted with physicians practicing in the province of Saskatchewan, Canada, during November 2020–July 2021. Thematic analysis identified core themes and subthemes. Seventeen physicians, including nine males and eight females, from eleven specialties completed the interviews. The pandemic brought to the forefront life’s temporality and a new appreciation for life, work, and each other. Most physicians found strength in values, such as gratitude, solidarity, and faith in human potential, to anchor them professionally and personally. A new need for personal fulfilment and hybrid care emerged. Negative feelings of anger, fear, uncertainty, and frustration were due to overwhelming pressures, while feelings of injustice and betrayal were caused by human or system failures. The physicians’ appreciation for life and family and their faith in humanity and science were the primary coping strategies used to build adaptation and overcome negative emotions. These reflections are summarized, and implications for prevention and resilience are discussed.

## 1. Introduction

Over the last decades, physicians worldwide have faced numerous professional pressures with implications for mental health struggles [1,2]. With an aging population afflicted by complex comorbidities, low physician-to-population ratio, and an ever-changing healthcare system, physicians’ difficulties have continued to grow in amount and in gravity [3]. The emergence of the SARS-CoV-2 pandemic in 2019 (COVID-19) instantly amplified the amount and severity of the pressures experienced. The high rate of mortality and disease transmission unveiled the shortcomings of many overstretched health systems, significantly impacting health care workers’ mental health and well-being [4]. Unsurprisingly, studies from across the world have documented an alarming increase in physical and mental health symptoms in physicians during the first and second pandemic waves [5,6,7,8,9,10,11,12]. For instance, one Canadian study demonstrated that physicians’ burnout increased by approximately 6% over the first year of the pandemic [12], An Italian study [13] found a high incidence of depression (57.9%), anxiety (65.2%), post-traumatic symptoms (55%), and burnout (25.61%) in healthcare workers [13]. A US national study revealed similar rates and demonstrated a positive association between physicians’ mental health symptoms and number of positive COVID-19 cases (e.g., more symptoms in physicians working in states with more than 20,000 positive cases or 1000 deaths compared with less affected states) [14]. The job-related stress was associated with an overloaded work environment where demand exceeded capacity. Likewise, in an 8-month longitudinal Japanese study, distress levels remained elevated even during periods of low infection, with a more significant impact seen during outbreaks [15]. Studies such as these suggest that prolonged and frequent exposures to high-stress situations can create lasting physical and mental health effects on physicians’ health and quality of life. The strong correlation between physicians’ psychological stress and mental health disturbances have been attributed to a number of factors, including resource insufficiency [16], a lack of social support, poor communication, and maladaptive coping strategies [17].

Besides the strain of caring for patients during the pandemic, physicians faced the uncertainty of becoming sick and dying themselves [7,18]. Consequently, the pandemic has forced physicians to face their fears and find ways to commit to their sense of duty [19]. This has not been an easy task. Given the tremendous external demands and pressures, many physicians did not realize the negative impact on their mental health and often failed to seek professional help and support [20]. Instead, they focused on developing coping skills meant to sustain themselves and help them navigate the adverse psychological burdens. These observations were corroborated in a large systematic review that captured the importance of building coping skills in health care workers to enhance resilience and well-being during the pandemic [21]. Still, as cases of infected patients, colleagues, and families increased, physicians’ ability to successfully bounce back from this pandemic trauma eventually declined over time [22]. In fact, more recent studies examining physicians’ experiences that were conducted during the first and second wave of the pandemic revealed that the positive effects of inter-personal factors (e.g., greater team bonding, respect and thankfulness from both colleagues and the public, and positive institutional dynamics and communication strategies) were short lived and quickly replaced by anxiety and panic in both their personal and professional domains [20,23].

Consequently, as time went by, it became evident that physicians’ understanding of the pandemic, their health challenges, coping strategies, and views were changing. Still, studies targeting these evolving experiences are missing. Since large epidemics/pandemics have become a major source of concern over the last decades, these observations seem timely to capture individual aspects to inform preventive and supportive measures for health care workers in general, and physicians in particular.

The current study aims to fill this gap by capturing the lived experiences of physicians from various specialties and locations (urban and rural). Data collection through individual interviews was conducted during the second and third waves of the pandemic in Canada when heavy lockdown restrictions were in effect and physicians were battling with different variants of the disease. Through the qualitative analysis, we explored physicians’ views of the pandemic, their individual challenges in professional and personal domains, their ways of interpreting and coping with these challenges, and their understanding of the implications. Several themes and subthemes emerged, capturing the crystalized beliefs and experiences of physicians from various specialties, including their discovered strengths worthy of reinforcing as well as vulnerabilities needing support.

## 2. Methodology

### 2.1. Study Design, Participant Recruitment, and Data Collection

An interpretive approach with the aspects of a phenomenological principle was used. In a phenomenological approach, the researcher seeks to understand the subjective experiences of participants [24]. To gain an understanding of the essence of physicians’ experiences, we invited all physicians practicing in Saskatchewan, Canada, to participate in a multi-phase study investigating physicians’ experiences during the COVID-19 pandemic. Information about the study was distributed to all the Provincial/Departmental Heads leading clinical departments in the College of Medicine at the University of Saskatchewan, who shared the study invitation to all physicians in their respective area. The study was also advertised in the Saskatchewan Medical Association newsletter. In Round 1 (R1), physicians completed an online anonymous survey of physical and mental health symptoms (e.g., depression, anxiety, and trauma-related disorders), coping strategies employed, and needed support (quantitative analysis). Physicians who completed R1 were invited to participate in Round 2 (R2), which entailed a confidential in-depth semi-structured interview by phone/video or in person (qualitative analysis). The current manuscript presents the results from the interviews conducted in R2. Of the 117 physicians who completed R1, 20 agreed to participate in R2. The Principal Investigator in collaboration with all authors developed the interview guide (Appendix A). The intention was to invite each participant to offer a narrative of their personal and professional experiences during the pandemic, their interpretations of these experiences, the coping strategies employed, and the lessons learned. All questions were open-ended questions, allowing participants to talk about their experiences freely in their own words. Saturation was met. All interviews were conducted and recorded by the PI and later transcribed verbatim for analysis. All participants provided informed consent prior to enrolment in the study. This study was conducted in accordance with the Declaration of Helsinki, and the protocol was approved by the University of Saskatchewan Behavioural Research Ethics Board (BEH 1953).

### 2.2. Analysis

Two researchers (CA and MRB) reviewed each transcript several times to fully understand each participant’s experience during the pandemic. NVivo 12 (https://www.qsrinternational.com/ first accessed on 3 June 2021) was used to facilitate the thematic analysis and presentation of the data. Using the 6-phase framework developed by Braun and Clark [25] (pp. 54–71), the researchers generated codes and extracted dominant themes. A general inductive approach guided the analysis, allowing for novel insights to emerge without the constraints imposed by a more structured deductive approach. The codes were compared and agreed upon during regular meetings. All themes and subthemes were reviewed and discussed with all authors, allowing for reflexivity and critiquing of our own and other team members’ interpretations to synthesize the final themes. This approach allowed the researchers to examine the richness of physicians’ lived experiences during the COVID-19 pandemic. We hoped to uncover the meaning-making process employed by physicians in dealing with their pandemic experiences, the range of successful or less successful coping strategies employed, and the interpretations and conclusions reached through this process.

## 3. Results

We contacted 20 potential participants who volunteered for the study, of whom we interviewed 17 (three participants did not respond despite multiple attempts to contact them). The demographic characteristics are presented in Table 1. The interviews lasted between 40 and 90 min and took place between November 2020 and July 2021. The transcription and qualitative analysis were conducted between September 2021 and March 2022. Data analysis of all interview responses revealed a few overarching themes and several subthemes. These themes and subthemes are grouped below according to the topic addressed by each of the seven questions of the interview.

### 3.1. Physician’s View of the Pandemic’s Origin

Figure 1 provides a glimpse into the most common words used by the physicians to depict their view of the origins of the COVID-19 pandemic.

#### 3.1.1. Natural Occurrence

Most of the participants believed the pandemic to be a natural or chance occurrence that was largely expected. This belief was linked to their knowledge of past pandemics and attributed to biological randomness. They perceived the pandemic as inevitable, although the magnitude of the COVID-19 pandemic surpassed most expectations. One participant shared, “*Pandemics have happened over the course of history and in my opinion, this just fits with that pattern …” (P15).* Another explained that *“there was 1918, 1957, then sometime in the 70s; then there was SARS and MERS and H1N1….” (P9).*

#### 3.1.2. Human Error

A smaller number of participants expressed the contributions of humanity or a combination of biological randomness enhanced by human actions—*“we seem to have fallen into this belief that we can continue …at great environmental costs and great expense to the ecology of the planet with no consequences*” (*P13*).

#### 3.1.3. Religious/Spiritual

Participants with strong spiritual or religious values incorporated this perspective into their overall interpretation of the pandemic’s origin. One participant commented that *“there has been human action…which you could call sin which has perpetuated the cause and spread” (P4).*

The word cloud presented in Figure 1 provides a visual representation of the most frequent words used by the physicians to describe their perceptions of the origin of the pandemic.

### 3.2. The Effects of the Pandemic on Physicians’ Experiences

#### 3.2.1. Facing Mortality and Developing Appreciation

In witnessing the tragedies caused by the pandemic while risking their lives to help patients, physicians faced their own mortality and examined their own priorities, as depicted by the quotes in Table 2. This acute awareness of life’s temporality revived their appreciation for being alive and reminded them of the things that mattered to them. From a personal perspective, most physicians commented on the importance of living a fulfilling life beyond their careers and articulated their renewed priorities. At the forefront, there seemed to be a renewed urgency for time spent with families and friends. Additionally, there was a renewed longing for pursuing postponed interests or forgotten hobbies. Younger physicians longed for holidays, while long-time practicing physicians saw the pandemic as an opportunity to explore retirement.

#### 3.2.2. Physicians Acknowledge Their Vulnerability

Several physicians lost someone to the virus, while the rest worried that this might happen. Personal and professional pressures augmented each other. When coupled with extreme work demands, some physicians struggled to find time for mental rest and self-care. An inability to balance work and personal life led to burnout and emotional suffering. The physicians who lived alone found it challenging to separate work from personal life. This challenge was even more difficult due to the loss of common everyday rituals, which typically provided structure and aided with daily tasks.

#### 3.2.3. Discovering Strengths

Even though deeply shaken by their vulnerability, most physicians were surprised to discover their own strengths and resilience, as well as a sense of appreciation and solidarity with their colleagues and the wider community. Problem solving and the ability to create new modalities of teaching students and treating patients were the sources of professional fulfilment and respect. A sense of resilience emerged as a result. The importance of good leadership was emphasized while raising the expectations to meet the heightened challenges—see Table 2.

#### 3.2.4. Mixed Feelings about Virtual Care

While all interviewed physicians recognized the benefits of virtual care, many preferred traditional methods of care delivery. Still, virtual care was seen as a progression into the future, offering sustainable and equitable treatment while providing easier access to patient care. The creation of billing codes for virtual care translated into a greater sense of financial stability and increased productivity for non-salaried physicians. Some discovered that virtual care could provide greater flexibility throughout the workday, allowing them to engage in multiple tasks.

In contrast, some physicians perceived virtual care as problematic for diagnosis and treatment. Moreover, with the restrictions on face-to-face visits, the participants estimated other systemic complications, such as longer wait times to see a specialist for needed in-person assessments and an unfortunate reduction in doctor–patient interactions and personal connection with patients.

### 3.3. Physicians’ Emotional Experiences Related to the Pandemic

Table 3 depicts the variety of emotions experienced and voiced by physicians practicing during the pandemic.

With no exceptions, all interviewed physicians were able to voice both positive and negative or mixed emotions resulting from their experiences of the pandemic. The negative emotions seemed to predominate for some, while the positive emotions were more voiced by others. Most seemed to agree on the heightened experiences of emotions, while very few were able to experience sustained peace. Throughout all interviews, the physicians often used a variety of words and expressions to express their positive emotions (e.g., contentment, peace, thankfulness, gratitude, and appreciation). Interestingly, negative emotions were strongly expressed with two key words, anger and frustration, as noted by 13/17 physicians in our sample (Figure 2).

#### 3.3.1. Empathy and Concern for Others

As some physicians reflected on the pain witnessed in their work and their personal lives, and their inability to help all those in need, they expressed heightened empathy and concern, and they acknowledged their increased sensitivity to the pain of others. At times, this motivated altruistic desires and behaviors, such as social activism, more generosity with their clinical time and efforts, and more understanding and support of family and friends. Unfortunately, at other times, it led to feelings of hopelessness, or even symptoms of burnout or depression.

#### 3.3.2. Anger and Frustration

Anger and frustration were the main negative emotion (see Figure 2) felt by the physicians in the sample and came from various sources. Most commonly, they were due to a perceived lack of congruency or directional exchange of information in an atmosphere where information was rapidly changing. High or unreasonable expectations from peers and/or administrators created confusion and misunderstandings. Adjusting to virtual care was difficult for some physicians, as traditional face-to-face care was perceived to be more effective for delivering personalized patient care and medical training.

#### 3.3.3. Guilt

Several physicians commented on a strong feeling of guilt related to the internal and external expectations of doing more and working harder. On the other hand, the physicians in leadership roles experienced some guilt about being removed from the hardest jobs and struggled to contribute more effectively or took on different roles that could support their colleagues who were adversely impacted by the pandemic.

#### 3.3.4. Injustice and Betrayal

Feelings of injustice and betrayal were particularly experienced by those who felt unsupported by leadership, at either the departmental level or health region/ministry level. The physicians with more stable income (e.g., salaried positions) felt safer and less affected than the physicians whose income dropped significantly due to the disparity between in-person and virtual care (e.g., those working in a fee-for-service model). In general, there was an expectation that those in authority should prioritize physicians’ health and safety, providing the necessary personal protective equipment and timely vaccinations. For some, these feelings of anger and frustration intertwined with their personal and family life, cultivating tension and guilt at home. For others, the pandemic removed the assurance of normalcy and routine and replaced it with a resigned acceptance.

#### 3.3.5. Anxiety and Fear

The initial shock of the pandemic’s rapid spread and the high mortality associated with it reverberated throughout the discourse of most physicians. This led to a sudden urgency to reflect on life’s temporality and fragility as it pertained to family members, friends, patients, and society at large. In addition, the continually evolving COVID-19 virus left the physicians uncertain about new approaches to care and availability of health care resources, and it created apprehension about the future.

#### 3.3.6. Gratitude

Gratitude was the most expressed positive emotion experienced by the physicians during the pandemic. It seemed to emerge from the contemplation of the surrounding danger and its contrast with the simple joys previously taken for granted. The physicians often felt fortunate and appreciative of all the professional and personal successes, as well as the small things in life, that suddenly gained value. On a personal level, they often described becoming appreciative of life’s small moments, refocusing their attention on emotional and physical health, spending more time with family, and aiming for self-actualization. On a professional level, there was great faith in human potential and appreciation for all the efforts witnessed in their brethren and worldwide.

#### 3.3.7. Pride

Pride was often an emotion experienced in the professional context, not only in response to the efforts and achievements experienced locally, but also in response to the global efforts that health care workers and scientists had engaged in to find solutions and to overcome this calamity.

#### 3.3.8. Togetherness

The sense of cohesion and closeness that developed through spending more time with each other in both personal and professional domains seemed to pervade all interviews. The physicians described a renewed satisfaction and appreciation for their collaboration and contributions to helping others as part of their work. Time spent with colleagues was often coined as a source of hope and joy. Extended time spent at work provided opportunities to join forces, socialize, and build friendships. Similarly, more time spent with family members “outside of the hustle and bustle of daily life” seemed to bring great satisfaction and intimacy.

The objective of the word cloud presented in Figure 2 is to visually represent the most frequent words used by the physicians to describe the range of emotions they experienced during the pandemic.

### 3.4. Physicians’ Coping

Table 4 summarizes the overarching themes, subthemes, and key quotes highlighting the coping strategies used by the physicians in order to adapt to the pandemic. The quotes depict responses to the question, “What keeps you hopeful during this time? Any reasons, values, goals, or beliefs?”

#### 3.4.1. Knowledge

Historical and scientific knowledge helped most physicians to contextualize and predict the course of the pandemic. Knowing that “pandemics are self-limited” anchored many physicians’ outlook on the crisis. Reflecting on the life cycle of similar events in the past, some physicians were optimistic that all past pandemics had ended within a few years, even during times with less medical advances.

#### 3.4.2. Problem Solving

Many physicians placed their hope in people’s ability to overcome similar obstacles through scientific discoveries and collaboration toward a single end goal. This mediator of hope fostered a desire for change through human actions and positive coping strategies, such as adaptation and resilience.

#### 3.4.3. Relationships

Togetherness and camaraderie offered great comfort and empowered many of the participants in the sample. Social support and connectedness with and through social networks positively influenced their psychological stamina and mental health during the pandemic. Even physicians who struggled with technology in their everyday lives pushed their boundaries to connect to family and friends using technology.

#### 3.4.4. Religion and Spirituality

Even though endorsed to a lesser extent, spirituality or religion-based value systems provided a source of identity and wisdom for several physicians. There was a sense of connectedness to a greater intelligence, and this feeling provided strength and daily purpose.

#### 3.4.5. Behavioral Coping

Most of the study participants used various behavioral activities to stay active while experiencing stressful realities. Many participants expressed faith “*in action*” and remaining “*in motion*” in order to cope and build resilience. Occasionally, they perceived a commitment to action to facilitate knowledge and build interpersonal relationships, endurance, self-efficacy, and positivity.

### 3.5. Lessons Learned and Their Implications

Table 5 summarizes the physicians’ perceived implications of the pandemic as they responded to the questions, “What lessons are you learning from the COVID pandemic? Specifically, to your professional life? How about personal life?”

Professionally, most physicians believed that going through the pandemic emphasized the need for preparation and for learning new ways that would allow them to adapt quickly to a potential new crisis. There was an acute awareness for a need for greater peer support, systemic changes, and efficient health care systems. The physicians noted the importance of effective communication, strengthening interdisciplinary workspaces, and advocacy as salient factors: “*I think we are better together as a health system, as uncomfortable sometimes it is, we can accomplish more together” (P1)*. As shown in Table 5, discovering new ways to improve emergency response and facilitating more efficient management skills were seen as essential priorities as part of the continuum of care. Many physicians believed that virtual care and good use of technology can offer great professional opportunities by facilitating knowledge transfer and access to health care. Many expressed that this is integral to strengthening/building healthcare resilience.

On a personal level, the participants highlighted the subtle shifts in their perceptions as they compared their experiences before and during the pandemic. The pandemic brought on an urgency to value time away from work. The pandemic confirmed and emphasized that physicians are as vulnerable as anyone to stress and burnout. One participant noted, *“physicians are people first and their profession second” (P1)*. At the time of the interview, almost half of the sample contemplated reducing work hours, taking extended time off work, or using virtual care to balance different facets of their lives. One physician who felt overworked and pressured later described feeling optimistic, having learnt the value of self-advocacy and learning to say “no” to additional tasks, despite wanting to do more.

All the participants acknowledged that the pandemic helped them gain a deeper appreciation for holidays and traveling. Many of them mentioned wanting to travel more once it was safe. They viewed their ability to visit family and friends as the best way to sustain interpersonal relationships and self-care: “*I am not taking for granted some things we had before such as travel to be closer to family” (P15)*. As many participants reflected, they recognized the importance of being compassionate toward themselves, their colleagues, and humanity alike.

### 3.6. Lessons to Be Learned

#### 3.6.1. Acknowledging Realities and Acting

The physicians voiced that humanity should be more aware of social realities, including the inequities in society that had been highlighted by the pandemic and the fragility of the health care system, as depicted in Table 5. Some physicians felt that humanity should learn that each community member is vulnerable to a crisis’s impact: “*People are fragile and we need to be kind to one another*”. The physicians also expressed that collaboration and working together are essential attributes of gaining compassion, overcoming obstacles, and preventing mental health difficulties. COVID-19 had revealed the interdependence throughout society, and the physicians expressed the hope that all these realities will become visible and will translate to a global strategy meant to build stronger relationships and resilience. A few physicians perceived that people have learned to value spending time with family and building better communities based on socially accountable values, consideration, and reciprocity.

#### 3.6.2. Strengths and Limitations

The strength of this study is the great variety of perspectives coming from 17 physicians from 11 specialties located across rural and urban healthcare settings, offering multiple insights and perspectives, with a good gender representation. The majority of the interviews took place at the point when vaccines had become accessible (shortly after the most debilitating phase of the pandemic), and this gave the participants an opportunity to reflect on a full range of experiences, including their strategies and emotions used to cope with the pandemic. The interviews were conducted by a psychiatrist, who is the principal investigator of the study, which gave some participants a therapeutic feel and comfort to reflect and speak openly about their pandemic experiences.

Every study has limitations in its study design. It is likely that our recruitment plan might have led to selection bias. Only physicians who were comfortable sharing their experiences and perceptions or felt the need to spare their time would have participated in the interview. The global impact and increased availability of knowledge about the pandemic could have influenced recollection or created vulnerability toward recall bias in relation to negative or positive events and feelings. Other factors such as social desirability might have influenced the reports collected in our interviews. As this portion of the study utilized a qualitative approach, the focus was placed on the richness and depth of experiences as interpreted by the physicians. Therefore, the small sample size limits the generalizability and transferability of said experiences as outlined in this manuscript.

## 4. Discussion

During the pandemic, physicians from all specialties have witnessed heart-breaking tragedies, faced tremendous pressures, and taken significant risks to perform their professional duties. These experiences have been emotionally very taxing, but also revolutionary. Taken together, all these factors have forced physicians to face their mortality, evaluate their professional and personal environments with great scrutiny, and prioritize accordingly. Similar to other studies, our participants displayed high levels of death anxiety [26,27]. However, these experiences motivated them to find solutions and discover new coping skills. Negative emotions of anxiety, frustration, and disappointment, and other mixed feelings were also observed in our study. In some instances, these emotions triggered greater psychological distress requiring psychiatric intervention, as was the case for one resident physician. However, the majority of participants were able to resort to pre-existing and newly discovered positive coping strategies as a means of counterbalance. Our findings bring evidence of constructive solutions in the face of adversity, which supports the psychosocial resilience of frontline physicians’ framework put forward by Banderjee and colleagues [17]. The participants in our sample spoke of coping both as a learnt process and an intrinsic part of their identity and took pride in their ability to overcome the adversities associated with the pandemic. The tremendous value of collaboration, trust, cohesiveness, and intimacy was remarked on by all, both professionally and personally. Feelings of gratitude and appreciation for simple joys, time spent with family, building and anchoring faith, as well as the freedom of traveling and connecting, were acknowledged as powerful protective factors for mental health and promoters of resilience. On the other hand, the lack of professional support, authoritarian leadership, and the lack of financial equity had aggravated the impact of existing pressures and eroded some physicians’ trust in their leaders and/or the governing structures. This finding is consistent with recent studies reiterating the importance of trust in maintaining agreeableness and compliance with government guidelines during the pandemic [28,29]. The results also have implications for constructive strategies meant to enhance resilience at a system level. For instance, streamlining information in a rapidly changing environment (using mobile apps, etc.) can reduce anxiety about the lack of communication. Additionally, gathering physicians regularly in order to provide opportunity for sharing experiences without judgement might also be conducive toward team building and collaboration. Lastly, developing self-care monitoring toolkits or offering regular wellness checks (every 4–6 months) might further enhance mental health awareness and facilitate resilience. As noted before [30,31], the pandemic is not only a public health threat but also a large economic threat. Therefore, our study findings can also be relevant to the system at large, by ensuring healthy collaboration and constructive problem solving in a humane and empathic environment.

## 5. Conclusions

As the pandemic continued to wage war against physicians and health care systems, it was paramount to offer physicians a reflective space to discover the impact of the pandemic on their mental health, coping strategies, strengths, weakness, and lessons learnt to inform effective interventions to build resilience to combat the next wave of adversities. The observations from our study offer great insight into the morale and coping strategies employed by physicians during the pandemic and the implications for the endurance or potential collapse of the health care force. By understanding individual and systemic protective factors as well as vulnerabilities, we can better design and implement preventive measures that are meant to decrease vulnerabilities to depression, anxiety, burnout, or suicide in at-risk professions. In our view, this detailed understanding can bring awareness and inform the development of robust interventions meant to support healthy coping and resilience necessary to facilitate physical and mental health endurance in the face of catastrophic challenges.

## 6. Contribution to Knowledge

### 6.1. What Does This Study Add to Existing Knowledge?

There is a general expectation that physicians will accept the risks associated with providing patient care; a notion that was put into practice during the COVID-19 pandemic. Yet, physicians’ personal and professional experiences and views have been minimally explored.

To our knowledge, this is the only qualitative study exploring the personal and professional experiences of physicians practicing in Canada in various specialties during the COVID-19 pandemic and one of the few to qualitatively investigate the coping strategies used by these health care workers.

The positive experiences, the newly discovered priorities, and the new outlook on life described by the physicians are acknowledged as powerful protective factors against mental health difficulties (particularly anxiety, depression, and burnout) and as promoters of resilience.

The negative experiences and emotions showcase the need for better emergency preparedness protocols accompanied by interventions and tools to support healthy coping and work–life balance.

### 6.2. What Are the Key Implications for Public Health Interventions, Practice, or Policy?

In keeping with the needed interventions for the health care crises, our data suggest that physicians indeed need to be supported personally and professionally. By doing so, we have the potential to better prepare physicians to manage the risks that come with the profession and better enable them to thrive in the face of danger through some of the following measures (although not restricted to these only):

Programs designed to enhance health recovery.

Shifts in personal and professional attitudes that are geared toward camaraderie, appreciation, and prioritization of work–life balance.

Work environments and leadership that are sensitive to physical and emotional needs, and willing to promote camaraderie, collaboration, and respect, rather than authoritative approaches that only think of the system but forget about the people that compose it.

## Figures and Tables

**Figure 1 ijerph-20-05259-f001:**
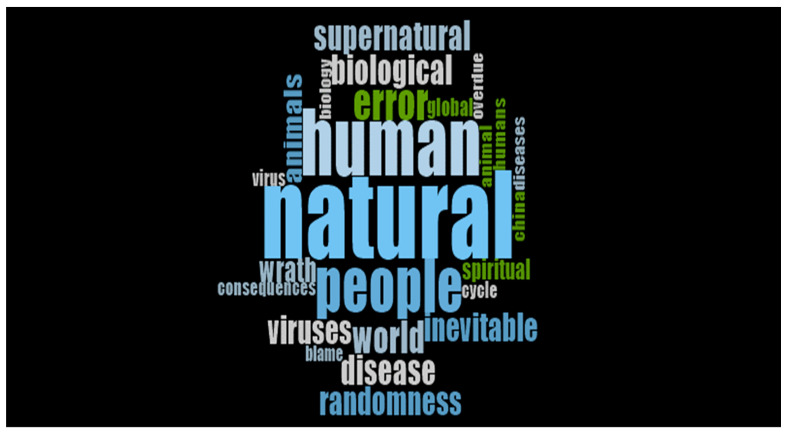
Word cloud depicting physicians’ views about the origin of the pandemic.

**Figure 2 ijerph-20-05259-f002:**
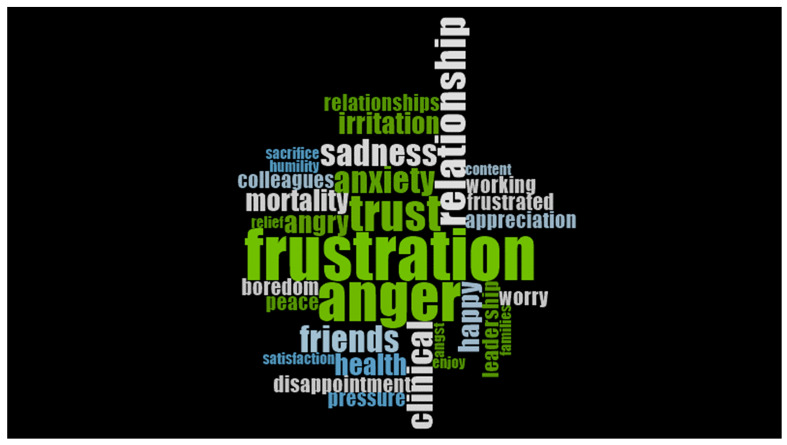
Word cloud depicting the most common words physicians used to express their emotions related to their pandemic experience.

**Table 1 ijerph-20-05259-t001:** Demographic characteristics of the 17 Canadian physicians interviewed during the COVID-19 pandemic.

Characteristics (N = 17)	n (%)
Sex	
Male	9 (53)
Age:	
25–44	5 (29)
45–65	10 (59)
66 and over	2 (12)
Marital status	
Married	15 (88)
Specialty or clinical role	
Family Medicine	5 (28)
Pathology	2 (12)
Urology	1 (6)
Pediatrician	1 (6)
Rheumatology	1 (6)
Obstetrics and Gynecology	1 (6)
Psychiatry	2 (12)
Anesthesiology	1 (6)
Ear Nose and Throat	1 (6)
Infection disease	1 (6)
Neurosurgery	1 (6)
Years in practice	
<1 to 5	4 (24)
6–10	2 (12)
11–15	4 (24)
16–20	2 (12)
>21	5 (28)

**Table 2 ijerph-20-05259-t002:** Overarching themes, subthemes, and associated quotes highlighted from the physicians’ experiences of the COVID-19 pandemic in response to the question, “How is the experience of the pandemic affecting you personally and professionally?”.

Theme	
Subtheme	Participants’ Quotes
RENEWED APPRECIATION FOR PERSONAL LIFE	
Facing mortality and developing appreciation for the time left Appreciation for family members and togetherness as a family	“In facing mortality, you discover that you want to spend time enjoying life…Just wanting to make time to commit to your interests and hobbies and basically just enjoying life not just working” (P13) “at my age… I can’t afford to ignore some of the other things I would like to be doing. If I want to get a horse and ride, I need to do that I can’t put it off indefinitely” (P13) “I won’t have them forever [parents] …I feel like I am given many chances to get that priority right…it is unfortunate it has taken this [the pandemic]’’ (P12) “So, my wife, kids, her parents, my dad and the immediate family, all of the energy that would have previously gone into socializing, hanging out with people that sort of stuff …the only priority right now is the family” (P7) “Just doing everything as a small family has been very lovely and at the beginning spending every evening with the kids that were home rather than everyone was running around. The usual family game night happening every night rather than once in a while. Eating together, learning to do more things together” (P6)
FACING VULNERABILITIES AND DISCOVERING STRENGTHS	
Acknowledging one’s vulnerability Discovering new skills Discovering new leaders Discovering resilience Adapting	“I have never dealt with burnout before, and I have been a little bit burnt out during the pandemic. I think it is the lack of being able to recharge in ways that I would normally recharge; that has really affected me and like I have not been able to travel to see my family or visit friends or you know do dinner parties or anything really … It has been humbling. I kind of thought I was immune to burnout, but I am obviously not” (P11) “There will be some moral distress and some baggage that will come out in time. We just need to be cognizant that there will be need for some help down the line” (P8) “So, this has forced all of us to learn a new skill set of how you conduct clinic virtually, be it by phone or be it online. So normally we can see the patient, we can tell they are not in respiratory distress, but you have to learn how to make those inferences through a phone visit, so it is kind of a new skill set that we had to learn” (P7) “It has created some leaders, not myself, but some young doctors I work with, have really been thrust into the leadership positions very early on in their career; so it has really helped foster a lot of personal or a lot of personal growth in those people” (P7) “I realized that we are not...just human beings, we are more resilient than we give ourselves credit for and even though we might have those feelings of guilt, we can adapt, and we can respond in positive ways” (P2) “With the support from the Saskatchewan Medical Association for virtual billing codes and support from our departments and our divisions we were able to adapt” (P2)
PERCEPTIONS OF VIRTUAL CARE	
Positive views on virtual care Negative views on virtual care	“COVID was the driving force that forced the government to open Pandora’s box and allow us to do virtual care …going forward virtual care is going to make a real difference for patients” (P7) “it is better for physicians, it is better for patients, it is better for access” (P10) “I really did not feel I was getting as full as an assessment…[it] delays care because I had to see all those people in person anyways. They all had to follow up with me in person” (P15) “it was very hard to start a practice when everything is virtual. You sort of miss the random encounters with people that will help you” (P16)

**Table 3 ijerph-20-05259-t003:** Themes and subthemes with associated quotes as the physicians shared the main emotions they experienced in their personal and professional lives during the COVID-19 pandemic.

Themes and Subthemes	Participants’ Quotes
POSITIVE EMOTIONS	
Peace and joy	“Um to be honest I think…it was all fun. Peace would be the closest one (emotion). I mean there was a job to do, and I was happy to do it” (P3)
Reflection	“It has created time and opportunity for contemplation around defining what is valuable …and then cutting out what is not valuable and then obviously trying to work on things that matter” (P8)
Gratitude and appreciation	“I feel very fortunate personally because I am in a situation where I am not losing my job, my income has not changed at all because I am salaried, I have gone through the pandemic and gone through the illness, and it has been very mild… could have been a lot worse” (P15) “I think there are lots of things in life that we at one point maybe just took for granted and now I think we have opportunities to just reflect upon how much joy we can get from simple things in life um and not to take things for granted” (P2)
Hope	“I have some hope because I see people working together a little bit more closely now and trying to solve problems that we do have” (P1)
Pride	“I feel a lot of pride especially in my colleagues because I think they are very tough, like it has been very hard and I have been so proud of all the people sharing the work with me... they have all stepped up and done the hard work that is necessary” (P4)
Togetherness	“It has been really nice to connect with them as friends [and] colleagues; we kind of unofficially socialize at work (laugh)” (P11) “There has been a sense of accomplishment and teamwork and interconnection that I had not previously experienced that has been in some ways a real joy” (P9) “I started doing shifts again and that made me very happy, to actually start doing shifts again and being happy to be back in the trenches with my friends and the other physicians” (P3) “our family has grown closer because we have had to spend more, you know, all of my free time...with my family, I think our family relationships are closer” (P4) “Loving time with my family, that has been really nice and spending time with my kids” (P11)
Relief	“Those meetings, I don’t drive so I had to travel through shuttle or taxis to different hospitals to attend the meetings… but now here they are all online on Webex, so that is also a feeling of relief” (P5)
MIXED EMOTIONS	“I think this has brought out the best and the worst in people and so whenever I think about the positive it is like I get a yang and right away I am thinking about the negative as well” (P12)
NEGATIVE EMOTIONS	
Empathy and concern for others	“Being relatively comfortable financially, having a house with good internet access, you look around and you see families in worse situations, and you wonder how they cope” (P6) “I feel like my empathy level has skyrocketed...I find it harder to create that clinical distance. I tear up a lot more at work, and I find it hard to hear somebody’s stories when I am on-call” (P11)
Anger, disappointment, and frustration	“It [the pandemic] has affected my wife’s mental and physical health. Seeing how it affected my kids was hard” (P6) “I may be more sensitive to some of the harms of the lockdowns” (P15) “I started to feel very irritated and angry because I felt that [those who] were more removed from clinical care weren’t understanding of the pressures…I felt that people were talking about irrelevant things” (P4) “We have a new area lead, who disappointed me greatly by being authoritarian when it was not necessary to be” (P17) “We are not even on the radar of the Saskatchewan Health Authority in terms of being worthy of a vaccine, so it went from stoicism and just getting the job done, to a little bit of anger, to like really over anger” (P7)
Guilt	“I struggle with guilt. About not being present enough for my kids or for my husband. I have to work more so I can’t do as many things as I used do with them, so I deal with a lot of guilt” (P4)
Injustice and betrayal	“We are not even remotely on the radar of the Saskatchewan Health Authority in terms of being worthy of a vaccine, so it went from stoicism and just get the job done to a little bit of anger, to like [really] anger.” (P7)
Anxiety and fear	“The worst thing is not knowing what is coming, the unknown, a good example would be the COVID variant [B117]; what are the implications of this in terms of the COVID vaccine, what are the implications...for healthcare resources.” (P2)
Self-doubt	“My income has reduced, and I am unable to travel…and not going out to meet friends and staying alone in the house is making me sort of question my purpose of life, and what I am doing” (P5)
Sadness	“Feeling sad, intimidated, unhappy not wanting to go to the job; but I love my job so much and I am so passionate if somebody asks me to come back during my vacation and do some work and even if they don’t pay me extra, I am very much willing to go, I like my job so much” (P5)

**Table 4 ijerph-20-05259-t004:** Overarching theme, subthemes, and key quotes highlighting the coping strategies used by the physicians to adapt to the COVID-19 pandemic.

Theme and Subthemes	Participants’ Quotes
MEDIATORS OF HOPE	
Faith in accumulated knowledge	“This too shall pass. This has happened before and there will be a price to be paid…it will be a better state sometime from now” (P8) “Pandemics have a natural cycle to them… of 2–3 years even if we didn’t have a vaccine. So as a scientist I know there is an end to this” (P7)
Faith in the ability to find solutions	“Humankind will have learned from this experience and that things will be better next time” (P4) “I believe that as people we do strive to come together in times of crises, and I think we will solve our problem one way or the other” (P1)
Faith in relationships	“My parents and my immediate family … phone calls or video calls to stay in touch (those have been helpful in terms of maintaining my own kind of mental wellness). If you express that vulnerability to close people around you, it actually can be quite powerful” (P2)
Faith in religion or spirituality	“I read scripture everyday…” (P4) “I spiritually feel this bad period is lasting more than ever like…. but I still feel that this period is not viable, it cannot stay” (P5)
Faith in action	“[motivator] to get out and go for runs and walks and bike rides in the summer and in the winter cross country skiing and snowshoeing” (P11)

**Table 5 ijerph-20-05259-t005:** Themes and subthemes highlighting the *physicians’ perceived implications of the pandemic* in response to the question, “what lessons are you learning from the COVID pandemic? Specifically, to your professional life? How about personal life?”.

Themes and Subthemes	Participants’ Quotes
IMPROVING WORK–LIFE BALANCE	
Reducing work hours Renegotiating contracts Redefining boundaries Making room for personal interest	“I was getting lost a bit in how much the job entailed and how many hours I had been working per week, so I need to pull away from that a little bit and give a little more time to myself” (P3) “…probably see if I can negotiate a slight change in my work so that there is more flexibility, or you know I might even go as far as see if there is a way to do a FTE reduction maybe go down to 0.9 or something just for the family side of things. Um but I guess the thing I am looking most forward to is life returning to normal so getting to do all of the things that you had to stop doing” (P7) “I hope that I have better work boundaries and that I [can] turn my phone off when I go home and not just constantly be available for work, I think that will be a big one for me if I could take that away from this” (P11) “I will go down to 60% and then maybe 25% I will just have to gage how long I stay working completely but I have a transition phase going on actively during the pandemic. Personally, it gives me time to gear up in other activities and prepare for having even more time to do those things” (P13)
PRIORITIZING TOGHETHERNESS	
Making plans to reconnect and socialize Prioritizing time spent with others in personal and professional circumstances	“I will honestly try to spend more time with my colleagues because I miss them…. take my wife to see her family” (P1) “Prioritizing spending time with people will be really high on our list and I think prioritizing um professionally those team building and then really getting those communication lines down especially professionally with our communities” (P14) “I am looking forward to house parties again, like dinner parties, like having people over for dinner…most important thing to do will be to see my family. I miss my parents and my brother, and I want to just go and meet and see them” (P11)
REJUVENATING HEALTH CARE	
Improving health care system and patient care Preparing for the future Improving continuity of care	“That we have improved our healthcare system. That we take better care of the more vulnerable” (P13) “That we do better pandemic preparation in the future that we build better systems as a result of this” (P13) “Find a better way of managing pandemics but still addressing other critical issues” (P15) “The importance of continuity care centers like having subspecialized care and like the COVID pandemic one of the side effects of it is it has been so having all the COVID stuff to deal with has meant that everything has been put on the back burner” P11
IMPROVING SOCIAL ACCOUNTABILITY	
Improving effort recognition Increasing community awareness Recognizing the importance of science	“The value of minimum wage workers and that minimum wage needs to go up” (P14) “Appreciating a big part of the greater community and we need to look out for each other” (P8) “don’t ignore science, don’t ignore the truth…. evidence matters, feelings aren’t facts...I hope they [people] step away from (hesitates) making decisions that are against [their] own self-interests, like not wearing a mask, like going to a big BBQ” (P10)

## Data Availability

The data presented in this study are available on request from the corresponding author. The data are not publicly available to maintain participants’ privacy and due to ethical reasons.

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
