# Peer review of "Doctors’ Professional and Personal Reflections: A Qualitative Exploration of Physicians’ Views and Coping during the COVID-19 Pandemic"

_ijerph, 2023, doi:10.3390/ijerph20075259_

Round 1

Reviewer 1 Report

 I appreciate the authors effort. However, the article needs a thorough improvisation. Few of them are highlighted below. 2)     Introduction: Needs to be revised. The introduction should be based on following format: 3)     The introduction should be better developed and organized to include, among others:- (one or) several paragraphs that position the paper compared to prior relevant academic literature.
-       (one or) several paragraphs that detail the contribution of the paper to the academic literature.
-       (one or) several paragraphs that summarize/present the study's results. This paragraph should be the ultimate paragraph in the introduction before giving the remaining parts of the paper.
-       The last paragraph of the introduction should "introduce" the remaining parts of the paper.
4)     All tables in the paper should be self-sufficient. Authors should add a caption for each table that explains its objective and content. The caption should include a brief description of the variables used in the table. Readers should be able to understand the table without going back to the text. Please proceed in the same way for the figures.5)     Literature Review: amount of literature and empirical evidence is relatively abundant, but introduction is poorly written. See few articles. No research hypotheses were formulated. 6)     Aysan, A., Kayani, F., & Kayani, U. N. (2020). The Chinese inward FDI and economic prospects amid COVID-19 crisis. Pakistan Journal of Commerce and Social Sciences, 14(4), 1088-1105.7)     Khan, M., Kayani, U. N., Khan, M., Mughal, K. S., & Haseeb, M. (2023). COVID-19 Pandemic & Financial Market Volatility; Evidence from GARCH Models. Journal of Risk and Financial Management, 16(1), 50.

Author Response

Please see updated attachment, with answers point by point

Reviewer 2 Report

This paper describes a study investigating the physician perspective during the COVID-19 pandemic. This is certainly a timely topic, and given physician rates of anxiety and depression are quite high, a very important one. 

Abstract is concise and well written. 

Introduction does provide a compelling argument for the importance of teh study. There are, however, some word choice errors, listed below-

Line 30-31 "likely to create vulnerability"; authors may have meant "like sharing mental healthy struggles vulnerably" 

Line 33-34- "have continued to aggravate"; authors may have meant "continue to grow in number and gravity"

The methods section is clearly written. A few points are missing- how were physicians contacted for these surveys? Were surveys anonymous, and was there any incentive to participate in the study versus not?

The results section is well-written and contains very interesting and intriguing data- the word clouds are excellent visual representations of the data, and the quotes in the tables complement the text well. One limitation of this study not mentioned is its size- certainly the geographic limitation is mentioned, but this is quite a small study and thus its results may be difficult to generalize. 

The discussion is also quite compelling although it would be helpful if the authors could provide concrete steps organizations or departments could do in order to help physicians through this current time. There are two grammatical issues as below-

Lines 361-362 "new coping"; authors may mean "new coping skills"

Line 364- likely needs a semicolon and not a comma

Conclusion is well-written. 

Reviewer 3 Report

The theme of the work is very interesting but being only a poor theoretical study remembering only a few studies, the work in this state, in my opinion, is not advisable to be published. The journal itself wants edifying results obtained by implementing innovative research methods that can lead to the identification of real solutions to current problems.

In order to achieve the objectives of such an invoice assumed in the title of this article, the authors use qualitative research and an exploratory survey conducted among experts, thus presenting us only the results of the R2 research. The structure presented, however, does not help us in identifying the real problems assumed from the proposed theme because sketching a pathology on the behavior of doctors due to COVID cannot be based on life opinions and experiences having only the structure of Table 1. Demographic characteristics of the participants (only one doctor is an infectious disease specialist).

The use of this method involves both obtaining answers to a series of questions and a free discussion, which gives the specialist the opportunity to express his own point of view, which is not clear from the presented article. Thus, medium- and long-term forecasts cannot be made of the evolution of this phenomenon, especially when there is very little information available.

A clear research model is not presented, just the results of an online questionnaire of 17 doctors in Saskatchewan, Canada.

I hope, this opinion will also give you inspiration and the opportunity to find a way to continue the theoretical study with an empirical one and to develop a new model, your own, based on predictions and using simulation and mathematics. You can do a comparative analysis by country/region/continent to develop a long-term strategy for a hospital.

In this form, this article cannot be published, because it has serious structuring deficiencies, further experiments are needed, and the research is not carried out correctly it must be deeper, otherwise, we cannot draw conclusions on a phenomenon, or behavior only by applying an online questionnaire.

Round 2

Reviewer 1 Report

Just look to the references part once again. As few mentioned references in the author's response letter are missing in the text file.   

Author Response

Thank you very much for your great suggestions. We are pleased the manuscript meets the reviewer's expectations in the current form. In order to address the last concern we have added two final sentences to the discussion where we cite the references suggested. 

The sentences read as follows:

"As noted before [31,32] the pandemic is not only a public health threat but also a large economic threat. Therefore, our study findings, can also be relevant to the system at large, by ensuring healthy collaboration and constructive problem-solving in a humane and empathic environment."

We hope these changes address the reviewer's final concern.

Reviewer 3 Report

This version of the work is much better grounded, explained starting from the theoretical context of the study up to the updating of the methodology and I hope to constitute a guide for specialists in the field.

Author Response

Thank you so  much for your great suggestions, and for your acceptance of the manuscript in the common form!